# A Massive Adenoma of the Uterine Tube in a Young Intact Female Dog: Surgical Intervention and Outcome

**DOI:** 10.3390/vetsci12030253

**Published:** 2025-03-07

**Authors:** Kazuyuki Terai, Ryou Tanaka, Aki Takeuchi, Kazumi Shimada, Miki Hirose, Aimi Yokoi, Ikki Mitsui, Lina Hamabe

**Affiliations:** 1Department of Veterinary Medicine, Faculty of Agriculture, Tokyo University of Agriculture and Technology, Fuchu 183-8509, Japan; fq9913@go.tuat.ac.jp (K.T.); fu0253@go.tuat.ac.jp (R.T.); fv5028@go.tuat.ac.jp (A.T.); kazumi-s@go.tuat.ac.jp (K.S.); s212240r@st.go.tuat.ac.jp (M.H.); s214816w@st.go.tuat.ac.jp (A.Y.); 2Laboratory of Veterinary Anatomy, Faculty of Veterinary Medicine, Okayama University of Science, Imabari 794-8555, Japan; i-mitsui@ous.ac.jp

**Keywords:** dog, oviduct, fallopian tube, neoplasia, adenoma

## Abstract

This is a report of a rare case of a uterine tube tumor in a young dog, a condition that is extremely rare in dogs. A one-year-old female Pomeranian was presented with a distended abdomen caused by a large abdominal mass. Despite its size being nearly 30% of the dog’s body weight, the tumor did not appear to cause any significant discomfort. Imaging studies revealed that the mass originated from the reproductive system, and then the mass and its affected reproductive organs were surgically removed. Histopathology identified the tumor to be a benign uterine tube adenoma, with no evidence of metastasis. The dog recovered well from the surgery, showing no complications or recurrence after more than two years. This case shows the possibility of young dogs developing reproductive tumors, underscoring the importance of regular veterinary checkups using imaging techniques. In this case, while pre-surgical identification of uterine tube tumor was challenging, complete surgical excision proved effective. Additionally, histopathology of seemingly cancerous tissue found during surgery was crucial for the diagnosis. The findings of this case contribute valuable information to the limited data on this rare condition in dogs, helping improve awareness and treatment approaches for similar cases in the future.

## 1. Introduction

In veterinary medicine, tumors originating from the uterine tube (synonym: tuba uterina) are commonly observed in poultry [1,2,3]; however, it is extremely rare in dogs, with only a few reports to date [4,5,6]. Reports on canine uterine tube tumors include epithelial tumors such as adenoma and adenocarcinoma, as well as mesenchymal tumors such as leiomyomas and lipomas [4,5,6]. These reports typically focus on their histopathological aspects, leaving clinical data and effective therapeutic options incomplete. The present report describes a case of uterine tube adenoma in a dog that was treated with surgical resection with a good prognosis, emphasizing its clinical presentation and histopathological characteristics, as well as the disease outcome.

## 2. Case Presentation

A one-year-and-two-month-old, intact female Pomeranian dog from a rescue organization was referred to the authors’ medical facility for the investigation of abdominal distension. The dog had been used for breeding and had a history of giving birth; however, specific details regarding the birth were unknown. According to the rescue organization, the dog had experienced one estrus before the operation. Clinically, at the time of presentation, the dog was in an anestrus stage. The dog had no history of abnormal estrous bleeding and had a normal estrous cycle. Upon the initial consultation, the patient showed no abnormality in its activity level, appetite, defecation, and urination, and there was no evidence of dyspnea due to abdominal distention. The dog weighed 3.9 kg with a body condition score of 3/5 and showed a severe abdominal distension but without any sign of pain or discomfort. Physical examination revealed a body temperature of 38.5 °C, with a heart rate of 144 beats per minute and a respiratory rate of 36 breaths per minute. The visible mucous membranes appeared pale pink, and no swelling or enlargement of the superficial lymph nodes was observed. Complete blood count (IDEXX ProCyte Dx; IDEXX Laboratories, Inc., Westbrook, ME, USA) revealed mild leukocytosis (22,800/μL; reference interval (RI) 2870–17,020/μL) and mild anemia (hematocrit: 30.0%; RI 30.3–52.3%). Serum biochemical analyses were performed using the (DRI-CHEM NX700; FUJIFILM Corporation, Tokyo, Japan), and the results (albumin; 2.7 g/dL, blood urea nitrogen; 11.7 mg/dL, alanine aminotransferase; 15 U/L, sodium; 139 mEq/L, potassium; 3.8 mEq/L, chlorine; 102 mEq/L) were within reference intervals. Thoracic radiography did not show any apparent abnormalities. Abdominal radiography revealed a left abdominal mass occupying the area from the last rib to the pelvis with displacement of the small intestine to the right, along with decreased contrast of intra-abdominal organs (Figure 1A,B). No other obvious abdominal abnormalities were noted. An abdominal ultrasound (ALOKA LISENDO880; Hitachi Ltd., Tokyo, Japan) detected a mass lesion extending throughout the left abdomen, with the mass located just ventral to the left kidney and in close contact. The left ovary and uterus were not identified around the left kidney. The mass exhibited heterogeneous echogenicity depending on the region, predominantly consisted of a homogeneous parenchymal area (Figure 2A), but partially contained a cystic area (Figure 2B). For the cystic area, anechoic regions were scattered within the cyst. Color Doppler examination revealed relatively poor vascularity in the parenchymal areas (Figure 2C) and abundant blood flow in the cystic areas (Figure 2D). A small amount of peritoneal effusion was observed around the bladder but was too small to collect for analysis. Echocardiography showed no obvious abnormalities in cardiac function. Based on these findings, a tentative diagnosis of an intra-abdominal mass near the left kidney suspected to originate from the female reproductive organ was made. Differential diagnoses included ovarian tumors (such as epithelial tumors, sex cord-stromal tumors, germ cell tumors, mesenchymal tumors) and uterine tumors (such as mesenchymal tumors, epithelial tumors).

Surgical excision of the mass was performed through median laparotomy. Subcutaneous injection of atropine sulfate (Atropine Sulfate Injection 0.5 mg; Mitsubishi Tanabe Pharma Co., Osaka, Japan, 0.05 mg/kg) and intravenous injections of midazolam hydrochloride (Dormicum injection 10 mg; Maruishi Pharmaceutical Co., Ltd., Osaka, Japan, 0.2 mg/kg) and butorphanol (Vetorphale; Meiji Animal Health Co., Ltd., Tokyo, Japan, 0.2 mg/kg) were used as pre-medications. Cefazolin sodium (Cefazolin Sodium injection 1 mg; Nichi-Iko Pharmaceutical Co., Ltd., Toyama City, Japan, 20 mg/kg) was administered intravenously as the perioperative antibiotic. General anesthesia was induced with propofol (Propofol intravenous injection 1%; Fresenius Kabi, Tokyo, Japan, 4 mg/kg IV). After endotracheal intubation, general anesthesia was maintained with a mixture of isoflurane (Isoflurane for animal use; MSD Animal Health, Rahway, NJ, USA) and oxygen. Although the mass was large, ventilation, oxygenation, and circulatory dynamics were stable during general anesthesia. The dog was positioned in dorsal recumbency. An abdominal incision was made from the xiphoid to the anterior pubis for optimal visualization of the mass. The mass extended from the caudal aspect of the liver to the cranial aspect of the pubis, compressing the spleen cranially. The tumor was partially adhered to the greater omentum, and upon separating the adhesions to inspect the dorsal aspect of the mass, it was found to extend into the retroperitoneal space, closely adhering to the left kidney. Macroscopically, the mass had regions that were red in color, with a soft consistency and a folded surface, as well as solid area which was enveloped by a capsule. A retroperitoneal incision was made to separate the mass from the left kidney. Although the mass and kidney were in close contact, adhesions were minimal, allowing for blunt dissection and easy separation of the mass. Subsequent dissection of the tissue surrounding the mass with an electrosurgical device revealed that the mass was continuous with the uterus. Due to the continuous connection between the mass and the left uterine horn (Figure 3A), a decision was made to perform a complete ovariohysterectomy along with the mass removal. The ovaries and uterus, containing the mass, were subsequently removed as per standard procedure. The excised tumor had a diameter of 30 cm and weighed 1.086 kg (Figure 3B), representing 28% of the dog’s body weight. No significant bleeding was observed during removal of the mass. After removal of the mass, an exploration of the abdominal cavity revealed numerous white to red, 1 to 2 mm nodules on the parietal peritoneum (Figure 3C), some of which were biopsied and submitted to histopathologic examination. No lymph node enlargement was observed in the abdominal cavity, and no obvious metastatic lesions were found in other organs. There were no apparent postoperative complications, and the dog was discharged on the third day after surgery. For pain management, butorphanol was administered intravenously at a dose of 0.2 mg/kg until discharge.

The excisional mass was fixed in 10% neutral buffered formalin immediately after biopsy. Samples were taken from both the folded region and the capsulated area of the mass and were then embedded in paraffin wax after 2 days of formalin fixation. Four-micrometer-thick sections were stained with hematoxylin and eosin (HE) for histopathological examination. The mass was composed of tubular and papillary proliferation of tall columnar epithelial cells without atypia or pleomorphism (Figure 4A), and no histological differences were observed between the different areas of the mass. The tumor cells had distinct cell boundaries, abundant amphophilic cytoplasm, a round-shaped nucleus with minimal anisokaryosis, and indistinct nucleolus. Mitoses were 0 to 1 per 10 high-power field with an ocular field number of 22. Cellular polarity was well maintained, and the stroma was composed of abundant fibrovascular tissue. The neoplastic epithelial cells had cilia on their apical surface (Figure 4B). The neoplastic glandular lumen was filled with transparent to mildly eosinophilic fluid. Vascular invasion by tumor cells was not observed, and the surgical excision of the tumor was judged complete. Based on these findings, the abdominal mass lesion was diagnosed as a uterine tube adenoma. The left ovary was effaced by the neoplastic mass. The right ovary and oviduct exhibited normal structure and histological organization (Figure 4C). The uterine endometrium was hyperplastic. The grossly observed nodules on the parietal peritoneum were devoid of neoplastic change and composed of fibrovascular proliferation and mild pleocellular inflammation. On postoperative day 912, the dog remained asymptomatic, with radiographic and ultrasonographic examinations showing no evidence of tumor recurrence.

## 3. Discussion

This case presents clinical data on a rare tumor originating from the uterine tube in a dog. A particularly notable feature of this case is that the tumor developed at a young age. Previous studies have reported that uterine tube tumors in dogs, similar to other tumor types, are generally observed in middle-aged and older individuals [4]. To date, the youngest documented case of a uterine tube tumor in a dog involved a 4-year-old Yorkshire terrier [5], making the present case the youngest on record. In birds, where uterine tube tumors are more frequently observed, their incidence is reported to increase with age, with occurrences in young individuals being exceptionally rare [3]. This trend of increasing incidence with age may also apply to canine uterine tube tumors, although it is important to accumulate more cases in the future.

While the exact etiology of uterine tube tumors in dogs remains unknown, studies in birds suggest potential involvement of increased lifetime ovulation frequency and elevated blood estrogen levels [7,8]. Given the young age of the present case, which suggests a relatively low number of ovulations, a genetic predisposition, and other environmental factors may also be implicated in the dog. Various primary fallopian tube tumors have been reported in humans, but their occurrence is considered rare [9,10,11,12,13]. Involvement of a BRCA germline mutation or a TP53 mutation has been reported as a factor in the development of these tumors [14,15]. In dogs, the influence of genetic background has not been investigated, and future studies are required to reveal the pathogenesis of canine fallopian tube tumors. In particular, the fact that this case was observed in a young dog may provide important information in the search for factors that contribute to the development of fallopian tube tumors in dogs. To the best of our knowledge, no previous reports of uterine tube tumors in dogs have described cases with a history of whelping. However, since this case had a history of whelping [4,5,6], further studies are warranted to investigate the potential causal relationship between whelping and the development of tubal tumors. Furthermore, although estrogen levels were not measured in this case, future investigations into hormonal profiles may provide insights into the pathogenesis of uterine tube tumors in dogs.

Another feature of this case is the size of the tumor. Previous reports have documented uterine tumors in Yorkshire Terriers with an 8 cm tumor and German Shepherds with a tumor weight of 2.27 kg [4,5]. In the present case, the tumor accounted for approximately 30% of the body weight, highlighting the potential for uterine tube tumors to reach a considerable size without obvious discomfort. This fact emphasizes the importance of periodical checkups of dogs using various diagnostic imaging, such as radiography and ultrasound.

In this case, of a benign uterine tube tumor, surgical removal of the tumor was an effective treatment even with the considerable size of the mass. Preoperative ultrasound revealed a tumor compressing the left kidney, raising suspicion that the tumor originated from the female reproductive organs. However, it was challenging to specify the exact origin of the tumor as the uterine tube. Similarly to the present case, ultrasonographic findings of ovarian and uterine tumors have also been reported to show a mixture of parenchymal components and cystic areas [16,17], making it difficult to differentiate ovarian, uterine, and uterine tube tumors by ultrasonographic findings alone. Consequently, preoperative ultrasound provided limited discrimination of the tissue origin. While preoperative computed tomography might aid in distinguishing the tissue origin, its utility could be limited in cases of extensive tumor expansion, as we believe that accurately visualizing the uterine tubes, which are approximately 1–3 mm in diameter and located in close proximity to the ovaries [18], is particularly challenging on computed tomography, especially in small breed dogs. Additionally, in the present case, a preoperative tissue biopsy was not performed due to a potential risk of intra-abdominal seeding of possible ovarian tumors [19]. Despite limited information on the origin of the mass, proactive excision of the mass was a reasonable choice, mainly due to the absence of imaging findings suggestive of metastasis. At the last follow-up on postoperative day 912, the dog was in good health with a favorable prognosis. Given the absence of prior reports on the prognosis of uterine tube tumors in dogs, accumulating additional cases is essential to better evaluate the effectiveness of surgical resection for this rare condition.

Uterine tube adenomas and adenocarcinomas have been indistinguishable by histology alone, so additional information such as tumor implantation is crucial to decide their level of malignancy [4]. In the present case, although peritoneal lesions were grossly present alongside the uterine tube tumor, no histological evidence of neoplastic dissemination was observed. This reminds us that it is crucial to perform histological examination on seemingly metastatic lesions found during the surgical procedure.

Though a neoplasm of the uterine tube is very rare, this should be included in differentials upon evaluation of female reproductive mass-like lesions, and surgical resection was considered an effective treatment. Further accumulation of cases is necessary to gain a comprehensive understanding of uterine tube tumors of domestic animals.

## 4. Conclusions

Uterine tube tumors in dogs, though rare, can develop at a young age and grow to large sizes without causing noticeable symptoms. This case emphasizes the importance of regular veterinary checkups and diagnostic imaging to detect such conditions early. Surgical resection, even for large tumors, appears to be an effective treatment, leading to a favorable prognosis, as shown by the absence of recurrence over two years. Further research and case accumulation are needed to better understand the etiology, optimal treatment strategies, and long-term outcomes of uterine tube tumors in dogs.

## Figures and Tables

**Figure 1 vetsci-12-00253-f001:**
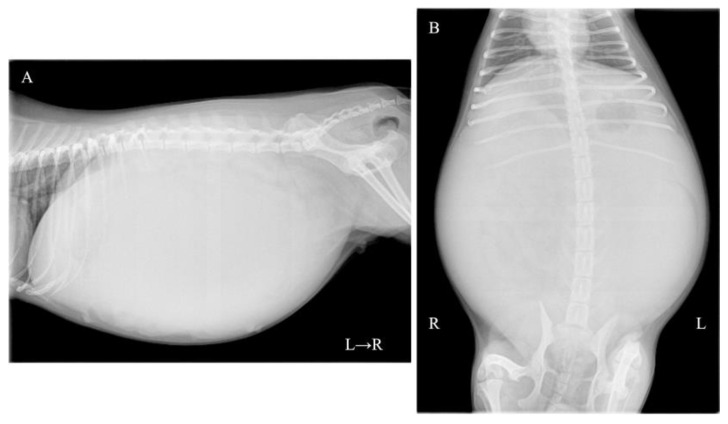
Abdominal radiographs: (**A**) right lateral recumbency image and (**B**) ventrodorsal image revealing a mass in the left abdominal region with displacement of the small intestine to the right.

**Figure 2 vetsci-12-00253-f002:**
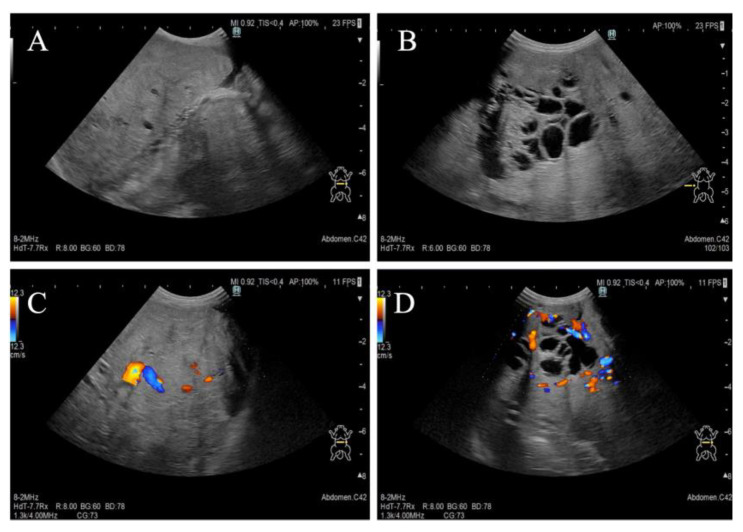
Preoperative abdominal ultrasound imaging. (**A**) Homogenous parenchymal area of the mass. (**B**) Cystic area with anechoic region. (**C**) Color Doppler imaging of parenchymal areas revealing relatively poor vascularity. (**D**) Color Doppler imaging of cystic area revealing abundant blood flow.

**Figure 3 vetsci-12-00253-f003:**
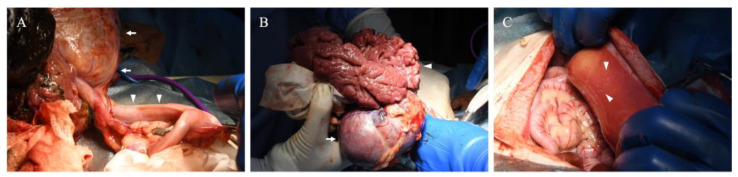
Intraoperative photographs. (**A**) The left uterine horn (arrowheads) was continuous from the mass (arrows). (**B**) Resected mass. The mass had a region exhibiting red in color, with soft consistency, and a folded surface (arrowhead), as well as solid area that was enveloped by a capsule (arrow). (**C**) Many white to red nodules (arrowheads) were seen on the parietal peritoneum.

**Figure 4 vetsci-12-00253-f004:**
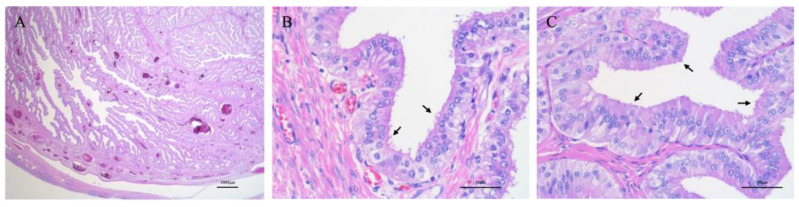
Histopathological findings stained with hematoxylin and eosin stain. (**A**) Photomicrograph of the mass (sub-gross picture). The mass is composed of tubular and papillary proliferation of tumor cells. Bar = 1000 μm. (**B**) Photomicrograph of the mass (high-power field). Tall columnar epithelial cells have cilia on their apical surface (arrows). No atypia is noted. Bar = 50 μm. (**C**) Photomicrograph of the contralateral normal uterine tube (high-power field). The epithelial cells are tall columnar and ciliated (arrows). Bar = 50 μm.

## Data Availability

Dataset available on request from the authors.

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
