# Peer review of "A Massive Adenoma of the Uterine Tube in a Young Intact Female Dog: Surgical Intervention and Outcome"

_vetsci, 2025, doi:10.3390/vetsci12030253_

Round 1

Reviewer 1 Report

Comments and Suggestions for Authors

I would like to suggest some points to be improved:
1.    The keywords must be corrected- better to choose only “Dog”, instead of “Tumor” to be “neoplasia” or “benign tumor”, I think is suitable to add “adenoma”.
2.    On row 46 as synonym of “uterine tube” my suggestion is to put in the brackets only the Latin name “tuba uterina”.
3.    On row 69 You must add the serum albumin results and to check on row 71 the number “102 mEq/L”.
4.    On row 99 is needed to add “median” or “midline” laparotomy.
5.    You said the neoplasia was with two different texture parts but nowhere I did not read from which part- cavernal or capsulated You took the samples. You must add that information around row 140. Did You open the parts with caverns? Was there any liquid? If “Yes”, did You take any samples for microbiological examination?
6.    In my opinion is necessary the CONCLUSIONS to be extended.
7.    REFERENCES
9% of the references are for a period of the last 5 years. It is mandatory to increase their number, You could add some information for adenoma for example.

Reviewer 2 Report

Comments and Suggestions for Authors

This report describes a rare case of uterine tube adenoma in a young Pomeranian dog. The abdominal mass was surgically removed, and the dog recovered well without complications or recurrence over the course of more than two years. This case underscores the importance of regular veterinary checkups and the use of imaging techniques, providing valuable insights into this uncommon reproductive tumor in dogs.

This review has been conducted appropriately; however, the following revisions are suggested:

  1. Line 60: The term “menstruation” is specific to humans or primates. It is unclear what you mean in this sentence—do you intend to refer to estrus or bleeding? Please clarify and provide more detail.
  2. Line 184: Birds are not the most suitable species for comparison with dogs, which are mammals. Please refer to the human literature, as numerous studies on adenomas in the reproductive tract are available.
  3. Line 186: The term “childbearing” is reserved for humans. Please revise accordingly.
  4. Lines 208-209: Kindly provide the reference for this statement.
  5. Lines 214-216: Please include the date of the last follow-up. While it is mentioned in the summary, it is absent in the main text. If possible, add more details in the "Case Presentation" section.

Observation: Please note that dorsal recumbency may not be the best position for dogs with large intra-abdominal masses, as it could lead to compression of the vena cava.

Additional Feedback:

  1. Main Question Addressed by the Research: This case report examines the treatment of a dog with uterine tube adenoma.
  2. Relevance and Originality of the Topic: To the best of my knowledge, prior to this paper, only one publication (describing two cases) has addressed the treatment of this type of tumor. The topic is both interesting and relevant, although it could benefit from more detailed information regarding the case (e.g., additional laboratory tests and follow-up data).
  3. Consistency of Conclusions with Evidence and Arguments: Yes, the conclusions are consistent with the evidence presented in the review.
  4. Appropriateness of References: The references should be improved, as mentioned earlier. It would be advisable to include additional sources to strengthen the review.
Comments on the Quality of English Language

Please, as mentioned in the corrections, the scientific language specific to human medicine should be replaced with that of veterinary medicine.

Reviewer 3 Report

Comments and Suggestions for Authors

This is an interesting case of a large cancerous tumor in a dog. The case is also interesting due to the fact that the tumor was diagnosed in a very young female dog.
The process of diagnosis and also the surgical procedure itself is presented.  In the discussion, the case was discussed with reference to other such cases.
The paper is interesting and should be published

What device was used to perform the blood tests?

What kind of usg camera was used for the examination?

Please add a description of pain management after surgery.

Reviewer 4 Report

Comments and Suggestions for Authors

The paper presented for review concerns a case of a fallopian tube tumor of spectacular size and occurring in an exceptionally young dog.
The paper is well and clearly written and presents an interesting case, worth noting in the literature.

I am in favor of accepting this paper for publication.

I am including a few comments below:

Line 60 how many estruses had this female until the operation

What was the stage of ovarian cycle- anoestrus (I suppose) , proestrus, estrus – it should be specified

Line 154 If “The left ovary was effaced by the neoplastic mass” couldn’t it be the source of tumor secondary infiltrating uterine tube?

Reviewer 5 Report

Comments and Suggestions for Authors

The authors describe a rare case of a uterine tube adenoma in a young animal. The study is interesting, and although this is not an entirely unique case, the approach is noteworthy and may serve as an important reference for clinicians encountering similar cases. Clinically and through imaging, it could provide valuable insights.

“No lesions were noted in the right ovary and oviduct.”

→ Please replace this with: “The right ovary and oviduct exhibited normal structure and histological organization.”

“Uterine tube adenomas and adenocarcinomas have been indistinguishable by histology alone, so additional information such as tumor implantation is crucial to decide their level of malignancy [4]. In the present case, although peritoneal lesions were grossly present alongside the uterine tube tumor, no histological evidence of neoplastic dissemination was observed. This reminds us that it is crucial to perform histological examination on seemingly metastatic lesions found during the surgical procedure.” This paragraph is unclear. Is histology diagnostic or not? If not, why were no additional techniques, such as immunohistochemistry, performed?

Please clarify that no evidence of metastasis was found, and mention explicitly that no lymph node enlargement was observed. Also, expand on the differential diagnoses.

Title: “Surgical treatment of a massive adenoma of the uterine tube in a young intact female dog” Would you consider changing the title? The value of this case report lies in the rarity of the tumor rather than the surgical procedure itself. The current title suggests that the novelty is in the surgical approach. While surgery is certainly relevant due to the tumor’s size, I wouldn’t highlight it in the title. Please propose an alternative title that better reflects the case’s significance.

Congratulations on your article!

Round 2

Reviewer 2 Report

Comments and Suggestions for Authors

Thank you for considering and accepting the changes suggested by this reviewer.